# An Assessment of Mortality among Elderly Brazilians from Alcohol Abuse Diseases: A Longitudinal Study from 1996 to 2019

**DOI:** 10.3390/ijerph192013467

**Published:** 2022-10-18

**Authors:** João de Sousa Pinheiro Barbosa, Leonardo Costa Pereira, Fabiana Xavier Cartaxo Salgado, Rodrigo Marques da Silva, Margô Gomes de Oliveira Karnikowski

**Affiliations:** 1Graduate Program in Health Sciences and Technologies, Faculty of Ceilândia, University of Brasilia, Metropolitan Center, Brasilia 72220-275, Brazil; 2Faculty of Education and Health Sciences—FACES—Medicine Course, University Center of Brasília—UniCEUB, 707/907, Asa Norte, Brasilia 70790-075, Brazil; 3Faculty of Health, Euroamerican University Center—UniEURO, Avenida das Nações, Trecho 0, Conjunto 05, Asa Sul, Brasilia 70200-001, Brazil

**Keywords:** elderly, mortality records, alcoholism, alcohol-related disorders

## Abstract

Alcohol use disorder (AUD) is a worldwide public health problem, being an important aggravating factor of comorbidities found in the elderly, with the potential to increase mortality indicators for this age group. Objective: To analyze alcohol-induced deaths in elderly people with alcohol-related disorder in Brazil between 1996 and 2019. Methods: An ecological study was conducted with secondary data obtained from the Brazilian Unified Health System (SIM) Mortality Information System from 1996 to 2019. TabNet/DATASUS, Excel^®^ 2016 and SPSS 21^®^ were used to prepare the results. Results: Between 1996 and 2019, 85,928 alcohol-induced deaths were recorded among the elderly (>60 years); in 1996, the lowest number of deaths was recorded (*n* = 1396), and in 2018, there were the highest number of deaths (*n* = 5667). In the profile of the elderly, there was a predominance of men (88%). Mortality from AUD was due to alcoholic liver disease (62.2%), followed by mental disorders due to alcohol use (37.3%). Conclusions: Coping with AUD is a public health problem that aims to reduce the number of deaths from diseases, conditions and injuries in which alcohol consumption is the causative agent, in addition to preventing deaths to which alcohol contributes.

## 1. Introduction

The elderly population tends to use health services more frequently than other age groups. The main causes of this demand for health services are related to the increase in neoplasms and external causes, as well as cerebro-cardiovascular, respiratory, and osteoarticular diseases and those related to the gastrointestinal system [1,2,3]. This standard of morbimortality for chronic illnesses in elderly people is also identified in other developing countries given that such clinical situations are aggravated by harmful alcohol consumption [1,2].

The condition of senescence is marked not only by the functional physical decline, generating physical dependence in the elderly, but also by cognitive decline, culminating in the reduction of conscientious decision making [3]. It is important to highlight that in recent years, there was an increase of deaths caused by mental and behavioral disorders, functional incapacity and falls. All these situations were aggravated by the extreme consumption of alcohol [3,4]. In a study conducted by Tuono et al. [5], there was an increase of 112% in the total number of deaths of elderly people who suffered from mental and behavioral disorders. It was found that 64.3% of deaths from mental disorders are due to the use of psychoactive substances, with a predominance of alcohol. According to the Brazilian Institute of Geography and Statistics (IBGE) through the Continuous National Household Sample Survey (Pnad Contínua) released in July 2022, the elderly population, that is, people over 60 years of age, represents 14.7% of the total population residing in Brazil, which represents in absolute numbers a total of 31.23 million elderly people [6].

The System of Information of Mortality (SIM) was developed and implemented in 1975 by the Brazilian Ministry of Health. This system allows the public consultation of the main causes of death, thus making it possible to know the epidemiologist profile of deaths in the entire country [7]. Mortality is an important health indicator that contains relevant information, which may reflect health conditions and the performance of health systems [8,9].

Harmful alcohol consumption is perceived as a behavior of young people, but in recent years, this behavior has been increasing among older adults. This study aims to analyze the causes of alcohol-induced death in chemically dependent elderly people between 1996 and 2019 in Brazil.

## 2. Materials and Methods

### 2.1. Study Design

This is a descriptive epidemiological study based on population data with ecological characteristics, in which the alcohol-induced disorders factor was observed between 1996 and 2019 in different georeferencing contexts, with stratification according to socioeconomic and cultural profile of elderly Brazilians [10].

The secondary data were collected from the electronic platform of the Ministry of Health, developed by the IT Department of the Brazilian Health System (DATASUS), by means of the System of Mortality of the Brazilian Health System, SIM/SUS.

To identify the sociodemographic profiles and the causes of mortality in elderly Brazilians due to abusive alcohol consumption, the experimental design procedure shown in Figure 1 was adopted.

### 2.2. Sample

The International Classification of Diseases ICD-10 was used [11] to determine deaths due to alcohol-related disorders in SIM/DATASUS/MS (www2.datasus.gov.br/ accessed on 10 December 2021 as the underlying or necessary cause of death. Therefore, in the present study, it was assumed that at the time of death, the individual should be 60 years old or older, of either sex, and from any region of the country, together with the information contained in the death certificate granted by the physician in charge. As exclusion criteria, those data that showed duality in the cause of death regarding alcohol abuse and other diseases were adopted.

### 2.3. Procedures

For the classification of the mortality conditions, the International Classification of Diseases - ICD10 of the World Health Organization WHO were used codes were used for death from illnesses caused by abusive alcohol consumption (Table 1). Here, the SIM base is guided by the Declaration of Death document, section IV, which identifies the conditions and cause of the death, which must be methodically filled out in items “a” to “d”, the terminal causa being described in item “d”, which was used in the present research [12].

To determine the mortality due to alcohol abuse in the elderly population and to compare it with the mortality of the non-elderly population, from 1996 to 2019, the mortality coefficients for both were calculated as represented in the equation below. The population data were based on the results provided by the Brazilian Institute of Geography and Statistics (IBGE) and resident population—study of population estimates by municipality, age and gender 2000-2020—BRAZIL (www2.datasus.gov.br/ accessed on 10 December 2020 [13,14]). Individuals aged 60 years or older were considered elderly.

To determine the mortality rate for alcohol-related disorders, two mathematical formulas were established, the first with the objective of calculating the mortality rate for people over 60 years of age (elderly) and the second formula for calculating the mortality rate in persons under the age of 60 years.

First formula: Coefficient of mortality of the elderly population for alcohol-induced disorders (CMiA) = number of deaths of elderly for alcohol consumption (ni) ÷ elderly population of (pi) × constant 100,000 inhabitants (c).

First formula Coefficient of mortality of the population younger than 60 years of age for Alcohol-Induced Disorders (CMA) = number of deaths for consumption alcohol of the total population (*n*) ÷ number of deaths in the population younger 60 years of age (p) × constant 100,000 inhabitants (c).

### 2.4. Statistical Analysis

For descriptive analysis, the DATASUS tools TabWin and TabNet Mortality Information System and Microsoft Office^®^ Excel 2016 were used to make graphs and tables, in addition to SPSS 21^®^. Categorical data are presented by absolute and/or relative frequency. Numerical variables are presented as mean ± standard deviation (SD). The Prais–Winsten regression model was used to analyze temporal trends, mortality coefficients from alcohol consumption, the number of deaths due to alcohol consumption (Y), and time (five-year period). Mortality trends were interpreted as increasing, decreasing, or stationary. The five-year coefficients of variation of the measures were evaluated and the confidence interval of 95% (CI95%) was estimated.

## 3. Results

From 1996 to 2019, 348,488 deaths due to alcohol-induced disorders in the country were registered in the Mortality Information System of the Ministry of Health of Brazil, considering all age groups. The number of deaths due to alcohol consumption in the country recorded for the elderly in the same period was 85,928, representing 24.7% of all deaths due to alcohol use disorder. The lowest number was recorded in 1996 (*n* = 1396), and in 2018 was the highest number of deaths for this age population (*n* = 5667).

The mortality coefficient for alcohol in the elderly was 11.3/100,000 elderly people in 1996, with a peak in 2012 from 21.2/100,000 elderly people to 19.3/100,000 elderly people in 2019 and with an average mortality coefficient of 17.6 ± 3.24/100,000 elderly inhabitants for the period from 1996 to 2019, while for people under 60 years of age, the average mortality coefficient was 6.5 ± 0.63/100,000 inhabitants.

The mortality coefficients from abusive alcohol consumption for the age groups under 60 years and for the elderly population of Brazil, corresponding to the period from 1996 to 2019, are represented in Figure 2.

In the elderly, the greatest cause of death due to alcohol consumption was alcoholic liver disease (K70) representing 62.2% (*n* = 53,475), followed by mental disorder due to alcohol use (F10) 37.3% (*n* = 32,053), accidental alcohol poisoning (X45); voluntary self-intoxication by alcohol (X65) and alcohol exposure poisoning (Y15) together accounted for 0.5% (*n* = 400) of death cases.

Mortality among the elderly population in Brazil related to alcohol consumption is described for each Brazilian macro-region in Table 2.

It was observed that people aged 60 to 69 died more from alcohol consumption than other age groups in the elderly population (Table 2).

Table 3 shows that regarding the sex and ethnicity of the elderly who died from alcohol-related causes, there was a predominance of males, self-reported white or mixed race, with low education and married (Table 3). The most deaths occurred in hospitals and other health services, followed by in domiciles. Table 4 shows the number of deaths and the evolution of the coefficients and trends in deaths of elderly people from alcohol consumption in macro-regions of Brazil, 1996–2019.

## 4. Discussion

In Brazil, more than 15,000 alcohol-related deaths are recorded per year, 1262 per month or 42 per day, all induced by excessive consumption of this substance, proving that chemical dependence generated by alcohol directly affects the health of this population. According to the World Health Organization (WHO) [15,16], 5.9% of the total deaths in the world were caused by the harmful use of alcohol, which represents 3.3 million deaths. These data disclose that alcohol-related disorders are responsible for 1 in every 20 deaths. In Europe, alcohol consumption is even higher, accounting for 2545 deaths per day [15,17]. In 2018, alcohol abuse was implicated in about 180,000 cases of cancer and 92,000 deaths from cancer developed by alcohol in Europe [18,19,20], which proves that alcohol-related disorders is a public health problem is several countries, whether these are developed or developing.

In a survey carried out by Chrystoja et al. (2021) together with the Pan American Organization (PAHO) and the World Health Organization between 2013 and 2015 in the countries of the Americas, more than 85,000 deaths per year caused exclusively by alcohol-related disorders were identified in the total population [21,22]. In the same period, alcohol consumption per capita in the Americas was 25% higher than the global average, with consumption of this substance being responsible for 64% of deaths of people under 60 years of age [21].

Brazil occupies the 53rd position among the countries that consume the most alcohol, representing an average consumption of 8.7 L annually by people over 15 years old [19,20], a consumption well above the world average, which is 6.2 L per year [15]. In Brazil, men consume on average 13 L and women approximately 2.4 L [17], which could clearly explain why the mortality from alcohol consumption found in this research was so much higher for males when compared with females. In relation to the population of 60 years of age and over, mortality from alcohol use disorders for the period studied was also higher among men (Table 3) for all regions of the country, with emphasis on the southeast followed by the northeast, which obtained the highest records of deaths attributable to alcohol consumption.

It has been observed that alcohol consumption is associated with preventable premature deaths; in a meta-analysis of 81 studies about the factors of the mortality of people with alcohol use disorder (AUD), it was identified that alcohol consumption is an important factor of morbimortality in men with AUD. The risk of death is as much as 3 times that of men who do not consume alcohol, and in women, the risk of death from alcohol consumption increases by 4.6 times. In individuals under the age of 40, the risk is 9 to 13 times between men and women in this age group [7,21]. Studies show that most deaths attributed to alcohol consumption occur in people aged 50 to 59 and are prevalent among men. With this, alcohol consumption and abuse is killing people in the prime of their lives and impacting family, social and economic cycles [18,23].

Longitudinal studies suggest few changes in alcohol consumption as people age; this decrease may be related to the spontaneous cessation of alcohol use among the elderly, health problems that limit access to alcohol, or financial stress (35.36%) [24,25]. For Rigler, alcohol abuse is a common problem, but the disorder is little recognized among the elderly. There is evidence that in most cases of abusive use of alcohol, the habit is developed throughout life. However, at least a third of the cases develop when entering the elderly stage, and, in both cases, it requires evaluation and interventions of physiological, psychological and social aspects [26,27].

Garcia and Freitas, in their descriptive study with data from the Brazilian National Health Survey (PNS), found that the gross mortality rate accumulated in the triennium for these causes was 9.60 per 100,000 inhabitants (total), with 17.35 per 100,000 men and 2.15 per 100,000 women. Liver diseases were the main causes, corresponding to 55.3% of all deaths that had alcohol consumption as a necessary cause [28].

Treatment for chemical dependency in Brazil is free for the entire Brazilian population and is guaranteed by the Unified Health System (SUS) as part of the National Mental Health Policy [29]. All people with this type of treatment need are referred to the Psychosocial Care Centers for Alcohol and Drugs (CAPS AD). These services are composed of teams formed by medical professionals, nurses, social workers, psychologists and other professionals who attend crisis situations as part of psychosocial rehabilitation processes [30]. CAPS AD is a service modality that works 24 h a day and serves all age groups with disorders caused by the harmful use of drugs such as alcohol [31,32,33].

On the other hand, it is important to highlight that the mortality presented in this study is just the “tip of an iceberg”, since the remaining mortality regarding alcohol consumption in which this was not a sufficient cause, which includes deaths from other causes linked to alcohol consumption, such as: violent deaths, accidents, cancer, tuberculosis, and pneumonia [34,35].

Considering the importance of public policies related to alcohol for health, security, and the international economy, the WHO carried out a study with the participation of several experts from nine countries in order to evaluate different policies on alcohol consumption [36,37,38]. A list of ten “best practices” was composed based on the following criteria: evidence of effectiveness, existence of scientific support, possibility of transposition to different cultures, and costs of implementation and support [36]. Five practices refer to (regulatory) alcohol control policies: (1) establishment (and inspection) of a minimum legal age for purchasing alcoholic beverages; (2) government monopoly of retail beverage sales; (3) restriction of hours or days of sale; (4) restriction on the density of alcohol sales points; and (5) creation of taxes on alcohol.

Four other practices are directly related to the control of drinking and driving: (1) reduction of the blood alcohol concentration limit allowed for driving; (2) administrative suspension of the license of drunk drivers; (3) establishment of sobriety checkpoints; and (4) a “zero tolerance” policy regarding drunk driving, registered in the license, for several years, for novice drivers [36,37,38].

## 5. Conclusions

Changes in alcohol marketing policies in Brazil and the dissemination of systematic anti-alcoholism policies in the various spheres of Brazilian society to raise awareness of confronting harmful consumption of alcohol are expected. In Brazil, there is a great influence of the alcoholic beverage industry, and it has been supported by free market values and social concepts. Increasingly, in order to achieve the goal of reducing the harmful consumption of alcohol in Brazil, measures regulated by the governments for its applicability in society and adoption of measures to confront the power of the alcohol industries are needed.

The adoption of systematic and comprehensive anti-alcohol measures in Brazilian cities is of great relevance to the fight against harmful alcohol consumption, which includes targeted enforcement measures. Confronting alcohol use as a public health problem aims to reduce the number of deaths caused by diseases, conditions, and injuries in which alcohol consumption is the main agent, in addition to preventing deaths, which alcohol contributes to its occurrence. This process of promotion and prevention will have positive impacts in reducing costs related to the cost of rehabilitation from alcohol consumption.

Deaths caused by alcohol-related disorders are preventable, but there is a need to intensify promotion, prevention, and rehabilitation in primary health care and in specialized health centers. We have public health policies for the application of integrality in the Brazilian community; however, access to services is still insufficient, either due to the lack of qualified health professionals for this type of care or material resources to provide progress in meeting the demands of the population with AUD.

### Limitation of the Study

In the present study, it was not possible to analyze the quantity related to some subcategories of CID-10 mentioned below due to a limitation of the DATASUS Mortality System, since this system does not provide them. The subcategories not analyzed were: Alcohol-induced Pseudo-Cushing syndrome (E24.4); Nervous system degeneration due to alcohol (G31.2); Alcoholic polyneuropathy (G62.1); Alcoholic myopathy (G72.1); Alcoholic cardiomyopathy (I42.6); Alcoholic gastritis (K29.2); Alcohol-induced acute pancreatitis (K85.2); Assistance provided to the mother for (suspected) injury to the fetus caused by maternal alcoholism (O35.4); Fetus and newborn affected by maternal alcohol use (P04.3) Fetal alcohol spectrum disorder (dimorphic) (Q86.0); Presence of alcohol in the blood (R78.0); Evidence of alcoholism determined by blood alcohol levels (Y90.0); Evidence of alcoholism determined by the level of intoxication (Y91.0).

## Figures and Tables

**Figure 1 ijerph-19-13467-f001:**
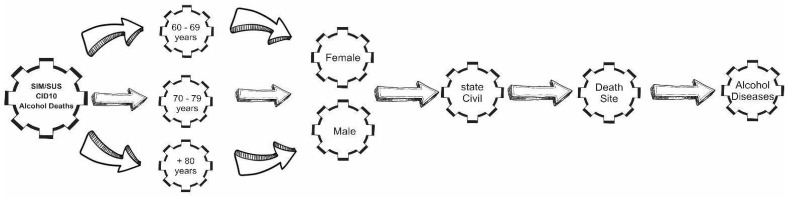
Flowchart of the data collected in SIM/SUS.

**Figure 2 ijerph-19-13467-f002:**
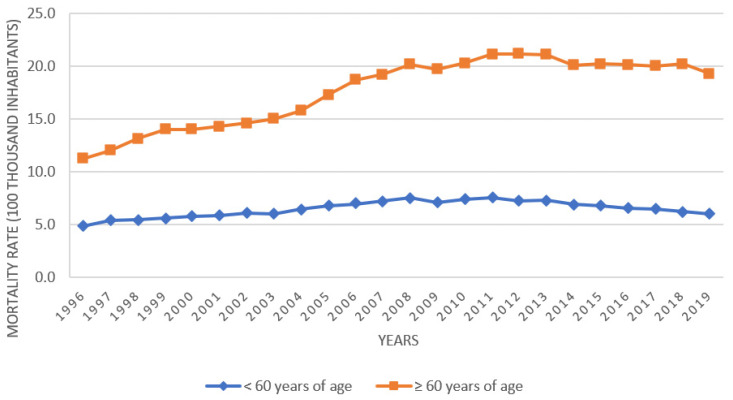
Evolution of the mortality coefficient from abusive alcohol consumption for age groups under 60 years and for the elderly population, from 1996 to 2019, in Brazil.

**Table 1 ijerph-19-13467-t001:** Classification of the conditions of mortality caused by alcohol abuse.

Causes of Mortality	CID.10 [11]
Illnesses of the liver	
Alcoholic illnesses of the liver	K70
Neuro-psychiatric Disorders	
Mental and behavioral disorders caused by alcohol consumption	F10.0
Nervous system degeneration due to alcohol	G31.2
Alcoholic polyneuropathy	G62.1
Poisoning/intoxication	
Accidental poisoning [intoxication] from alcohol exposure	X45.0
Voluntary alcohol self-intoxication	X65.0
Accidental poisoning [intoxication] from alcohol exposure, not determined	Y15.0
Evidence of alcoholism determined by alcoholemia levels	Y90.0
Evidence of alcoholism determined by the level of intoxication	Y91.0
Other illnesses (all other codes)	
Alcohol-induced Pseudo-Cushing Syndrome	E24.4
Alcoholic myopathy	G72.1
Alcoholic cardiomyopathy	I42.6
Alcoholic Gastritis	K29.2
Acute Alcohol-induced pancreatitis	K85.2
Alcohol-induced chronic pancreatitis	K86.0
Assistance provided to the mother for (suspected) injury to the fetus maternal alcoholism	O35.4
Fetus and newborn affected by maternal alcohol use	P04.3
Fetal alcohol spectrum disorder (dimorphic)	Q86.0
Presence of alcohol in the blood	R78.0

**Table 2 ijerph-19-13467-t002:** Mortality by CID-10 category related to alcohol consumption in the elderly by Brazilian region from 1996 to 2019.

REGION	CAUSE		AGE GROUP		
			60–69	70–80	+80
North	F10—Mental and Behavioral Disorders due to Alcohol Use		635	323	113
	K70—Alcoholic Liver Disease		1.203	594	270
	X45—Accidental Poisoning (intoxication) By and Exposure to Alcohol		6	3	1
	X65—Voluntary Alcohol Self-intoxication		6	1	1
	Y15—Poisoning [intoxication] by and exposure to alcohol, intent not determined		3	4	1
Northeast	F10—Mental and Behavioral Disorders due to Alcohol Use		6.013	2.896	1.298
	K70—Alcoholic Liver Disease		8.819	4.181	1.486
	X45—Accidental Poisoning (intoxication) By and Exposure to Alcohol		15	2	1
	X65—Voluntary Alcohol Self-intoxication		33	10	9
	Y15—Poisoning [intoxication] by and exposure to alcohol, intent not determined		58	30	14
Southeast	F10—Mental and Behavioral Disorders due to Alcohol Use		8.365	3.119	877
	K70—Alcoholic Liver Disease		15.497	5.843	1.378
	X45—Accidental Poisoning (intoxication) By and Exposure to Alcohol		26	5	2
	X65—Voluntary Alcohol Self-intoxication		17	6	3
	Y15—Poisoning [intoxication] by and exposure to alcohol, intent not determined		33	10	5
South	F10—Mental and Behavioral Disorders due to Alcohol Use		3.937	1.530	455
	K70—Alcoholic Liver Disease		7.085	2.858	631
	X45—Accidental Poisoning (intoxication) By and Exposure to Alcohol		10	3	0
	X65—Voluntary Alcohol Self-intoxication		18	2	1
	Y15—Poisoning [intoxication] by and exposure to alcohol, intent not determined		13	5	3
MidWest	F10—Mental and Behavioral Disorders due to Alcohol Use		1.625	650	217
	K70—Alcoholic Liver Disease		2.402	956	272
	X45—Accidental Poisoning (intoxication) By and Exposure to Alcohol		15	4	-
	X65—Voluntary Alcohol Self-intoxication		6	1	-
	Y15—Poisoning [intoxication] by and exposure to alcohol, intent not determined		7	7	-
	Total	(%)	55,847	23,043	7038
		(N)	65%	26.8%	8.2%

**Table 3 ijerph-19-13467-t003:** Profile of the elderly with a record of death due to alcohol consumption in SIM/DATA SUS between 1996 and 2019 in Brazil.

PROFILE	REGION					TOTAL
	NORTH	NORTHEAST	SOUTHEAST	SOUTH	MID WEST	
Sex *n* (%)						
Female	358 (3.69)	3.327 (34.31)	3.790 (39.08)	1.537 (15.84)	685 (7.06)	9.697 (11.29)
Male	2.809 (3.68)	21.538 (28.25)	31.393 (41.18)	15.015 (19.7)	5.476 (7.18)	76.231 (88.71)
Breed *n* (%)						
White	491 (1.26)	4.405 (11.32)	19.197 (49.31)	12.767 (32.8)	2.069 (5.31)	3.8929 (45.3)
Black	289 (3.61)	2.745 (34.33)	3.427 (42.85)	953 (11.92)	583 (7.29)	7.997 (9.31)
Asian	13 (2.84)	108 (23.58)	251 (54.8)	55 (12.01)	31 (6.77)	458 (0.53)
Brown	2.147 (7.04)	14.592 (47.86)	9.144 (29.99)	1.595 (5.23)	3.014 (9.88)	30.492 (35.49)
Indigenous	60 (22.47)	51 (19.1)	26 (9.74)	35 (13.11)	95 (35.58)	267 (0.31)
Not declared	168 (2.16)	2.964 (38.07)	3.139 (40.32)	1.145 (14.71)	369 (4.74)	7.785 (9.06)
Schooling *n* (%)						
None	1.024 (5.7)	8.716 (48.52)	4.161 (23.16)	2.470 (13.75)	1.593 (8.87)	17.964 (20.91)
1 to 3 years	757 (3.73)	5049 (24.91)	8.481 (41.84)	4.484 (22.12)	1.499 (7.4)	20.270 (23.59)
4 to 7 years	448 (3.29)	2.364 (17.34)	6.355 (46.61)	3.430 (25.16)	1.037 (7.61)	13.634 (15.87)
1 to 8 years	248 (3.87)	1.250 (19.53)	3.052 (47.68)	1.357 (21.2)	494 (7.72)	6.401 (7.45)
9 to 11 years	74 (2.96)	445 (17.77)	1.199 (47.88)	565 (22.56)	221 (8.83)	2.504 (2.91)
12 years or more	5 (3.57)	20 (14.29)	73 (52.14)	35 (25)	7 (5)	140 (0.16)
Not declared	636 (2.54)	6.956 (27.81)	11.860 (47.41)	4.228 (16.9)	1.335 (5.34)	25.015 (29.11)
Marital Status *n* (%)						
Single	1.193 (3.23)	1.0242 (27.72)	15.835 (42.86)	7.568 (20.48)	2.111 (5.71)	36.949 (43)
Married	408 (3.19)	3.213 (25.1)	5.446 (42.54)	2.804 (21.9)	930 (7.27)	1 2801 (14.9)
Widower	147 (1.91)	1.053 (13.7)	3.892 (50.62)	1.922 (25)	674 (8.77)	7.688 (8.95)
Legally separated	135 (7.86)	641 (37.33)	488 (28.42)	307 (17.88)	146 (8.5)	1.717 (2)
Other	325 (4.61)	2.695 (38.26)	2.409 (34.2)	964 (13.69)	650 (9.23)	7.043 (8.2)
Not declared	1.193 (3.23)	10.242 (27.72)	15.835 (42.86)	7.568 (20.48)	2.111 (5.71)	36.949 (43)
Place of death *n* (%)						
Hospital/Health Service	2.013 (3.36)	14.231 (23.73)	28.214 (47.04)	11.444 (19.08)	4.078 (6.8)	59.980 (69.8)
Domicile	1.001 (4.35)	9.781 (42.53)	6.130 (26.66)	4.326 (18.81)	1.758 (7.64)	22.996 (26.76)
Public highway	73 (6.56)	401 (36.06)	276 (24.82)	234 (21.04)	128 (11.51)	1.112 (1.29)
Other	69 (4.45)	382 (24.65)	521 (33.61)	395 (25.48)	183 (11.81)	1.550 (1.8)
Not declared	8 (2.76)	70 (24.14)	45 (15.52)	152 (52.41)	15 (5.17)	290 (0.34)
Total	3.164	24.865	35.186	16.551	6.162	85.928

**Table 4 ijerph-19-13467-t004:** Number of deaths, evolution of the coefficient and trend of deaths of elderly people from alcohol consumption in macro-regions, Brazil, 1996–2019.

VARIABLES	1996–2000	2001–2004	2005–2009	2010–2014	2015–2019	TOTAL	FIVE-YEAR VARIATION COEFFICIENT (%)	IC_95%_ *	TREND
	N	N	N	N	N	N			
North	186	252	581	947	1.198	3.164	73.0	4.13; 5.22	Increscent
Northeast	1.760	2.181	5.577	7.158	8.189	24.865	85.7	15.3; 18.7	Increscent
Southeast	3.933	4.292	7.080	9.234	10.647	35.186	99.8	8.7; 10.3	Increscent
South	2.067	2.153	3.406	4.229	4.696	16.551	90.1	39.4; 47.6	Increscent
MidWest	484	672	1.077	1.704	2.225	6.162	105.0	25.7; 30.2	Increscent
Total	8.430	9.550	17.721	23.272	26.955	85.928			

* IC_95%_: confidence interval of 95%.

## Data Availability

All data are presented in the article.

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
