# Peer review of "An Assessment of Mortality among Elderly Brazilians from Alcohol Abuse Diseases: A Longitudinal Study from 1996 to 2019"

_ijerph, 2022, doi:10.3390/ijerph192013467_

Round 1

Reviewer 1 Report

The paper is aimed to analyze alcohol‐induced deaths in elderly people with alcohol‐related disorder in Brazil and report some interesting data. The presentation of data is detailed, but there are some concerns to be clarified and resolved.

Lines 41-43- The sentence “ It is important to highlight that in recent years there was an increase of deaths caused by mental and behavioral disorders, functional incapacity and falls.” I crucial and should be outlined and discussed also in the discussion section

Lines 52-55         this is a repetition. Insert the bibliography in the previous paragraph and delete this.

Line 65                 what exactly means: chemically dependent elderly people

Lines 87-89         This exclusion criterion is very restrictive and potentially misleading. In fact, as reported also in lines 138-140,  alcohol is a potentiating and aggravating factor in many chronic diseases and therefore excluding the dual diagnosis potentially excludes many subjects who have suffered harm from alcohol use. This choice could also explain the great variability in annual prevalence, which cannot be explained by external factors in alcohol behavior and use.

line 110               of what ?? Please complete the sentence

line 162               in table 2 mortality is reported in absolute numbers. To facilitate and understand the figures, it would also be necessary to enter the prevalence with respect to the youth population or the total population (for example the% of deaths by age group

Line 165              In table 3 the term "Skin Color" is not scientifically acceptable. This specification in my opinion should be deleted, with its references in the text.   The table also indicates a percentage. What are you referring to? should be clarified in the legend.

Discussion section.

The discussion is very complex, but to make it more readable it is necessary to shorten it. The practices to be recommended to decision makers should be underlined and highlighted, for example in a separate figure or box.

Author Response

Line 41 – 43 – belongs to the introduction of the study, we made a comparison that elderly alcohol users die more from mental disorders, functional disability and falls than elderly people who do not have alcohol use disorder.

Lines 52 – 55 – paragraph removed and bibliography added as suggested.

Line 65 – Substance use disorder according to DSM-V has its subdivision as alcohol use disorder. Chemical dependent elderly would be the elderly dependent on several drugs, in this study we limited it to elderly people with alcohol use disorder. With the suggestion we changed the term in the study to: Alcohol-Induced Disorders

Lines 87 to 89 – The death certificate in Brazil is an official document issued by the Ministry of Health to all Brazilian health institutions, whether private or public throughout Brazilian territory, as a criterion for completing the death certificate the medical professional who diagnoses the death of the patient takes into account this rule, which is established by federal law 11,976 of July 7, 2009. As a result, the Mortality Information System only registers one International Classification of disease (ICD 10), for example: Mr. Brazilian died and the doctor identified the cause of death of Mr. João on account of alcohol in the death certificate will be registered as CID. F10 - Mental and behavioral disorders due to alcohol use. In line 314, the limitation of the study was added, which exactly justifies this doubt of the reviewer.

Line 110 – Correction performed.

Line 162 - table number 2 addresses the cause according to ICD, age group, Brazilian region and elderly group (60 - 69; 70 - 80; +80 years old) these data are worked in absolute numbers, but at the end of the table the total number of deaths by age group is in %, there is no need to put each cause of death from Alcohol-Induced Disorders as a percentage.

Line 165 – we will change to race to characterize the research subject's ethnicity. 

Discussion section – we relate the study to the main scientific evidence on the subject studied, it is a survey where it is little addressed in the databases, so we need to deepen the discussion related to the subject studied. To reduce the discussion, we will miss important points and we would avoid the objective of the study.

Reviewer 2 Report

Review:

Title: Evaluation of Mortality of Elderly People Brazilians due to Alcohol Abuse: A Longitudinal Study from 1996 to 2019

The manuscript shows very important problem, but it has to be corrected and changed:

Line 2: My proposition of the title change: Evaluation of Mortality of Elderly People Brazilians due to Alcohol Abuse illnesses caused by abusive alcohol consumption: A Longitudinal Study from 1996 to 2019, because of the authors did not identify diagnosis of alcohol dependence or alcohol abuse in SIM, but only   „illnesses caused by abusive alcohol consumption”.

Line 14: In the abstract - authors used acronym „ART”,  but in the text of the manuscript authors use different acronyms for example AUD. Authors should use one type an acronym. 

Line 44: „112%” Is it correct?

Line 65: unify of a terminology: in the aim of study we see „in chemically dependent elderly people”, and in abstract „AlcoholRelated Disorder (ART)” (line 14) or in the text „abusive alcohol consumption” (line 77) or „mortality coefficients from alcohol consumption” – I think that is adequate term/definition (lines 124-125), „alcohol use disorder (AUD)” (line 171) and etc. Authors have to change term/definitione, which is compatible with SIM.

Figure 1: it is necessary to enlarge the letters.

In the results:

it would be better when authors will put description of tables (figure) under the tables (also figure).

In the discussion:

Lines 178, 182, 194: authors should change „alcoholism” for „alcohol use disorder (AUD)” or others terms.

Line 224 and 226: authors showed data for example (n=100 252) and (n=85,928), with comma and without comma. Authors should unify this.

Line 232: „… financial stress (35.36%)” (lack of references source).

Lines 266, 268, 269: lack of a spaces in front of a brackets.

At the end (in part of discussion and conclusion) the authors should describes more information about the practical intervention for older persons with AUD for example – about proposal prevention and treatment methods for this people group.

Author Response

• Line 2 - we will change the work title following the evaluator's suggestion, the title will be: Assessment of Mortality of Elderly Brazilians from Alcohol Abuse Diseases Caused by Alcohol Abusive Consumption: A Longitudinal Study from 1996 to 2019
• Line 14 – We standardized the acronym Alcohol-Induced Disorders (AUD) throughout the text, removing all the acronyms ART.
• Line 44 – this data is correct. It was a Brazilian study that investigated 7,332 deaths of women aged 10 to 49 years, residing in Brazilian capitals, which occurred in the first half of 2002, using the Reproductive Age Mortality Survey (RAMOS) methodology, through home interviews, information from medical records and autopsy reports. The study allowed for an increase (112%) of Mental Disorder as a cause of death when comparing the original statements and those redone after the investigation. Cases in which MD were referred, either as an underlying or associated cause, were analyzed. Study available: http://scielo.iec.gov.br/scielo.php?script=sci_abstract&pid=S1679-49742007000200003&lng=pt&nrm=is
• Line 65 – We standardized the terminology: alcohol-related disorders (AUD)
• Lines 77-78 - We standardized the terminology: alcohol-related disorders (AUD)
• Lines 124 – 125 - We standardized the terminology: alcohol-related disorders (AUD)
• Lines 171 – 172 - We standardized the terminology: alcohol-related disorders (AUD)
• Figure 1 – we enlarged the font to 10
• Lines 179, 182, 194 - We standardized the terminology: alcohol-related disorders (AUD)
• Lines 224 and 226 – the data was standardized by adding a comma.
• Line 232 – added bibliographic reference as suggested by the reviewer.
• Lines 266, 268, 269: spaces were added in front of the square brackets.
• End of study (conclusion) – added a paragraph featuring the reviewer's suggestion.
